# Evaluating digital literacy of health professionals in Ethiopian health sectors: A systematic review and meta-analysis

Alex Ayenew Chereka[1]*, Agmasie Damtew Walle[1], Sisay Yitayih Kassie[1], Adamu Ambachew Shibabaw[1], Fikadu Wake Butta[1], Addisalem Workie Demsash[1], Mekonnen Kenate Hunde[2], Abiy Tassew Dubale[1], Teshome Bekana[3], Gemeda Wakgari Kitil[4], Milkias Dugassa Emanu[5], Mathias Nega Tadesse[6]

1 Department of Health Informatics, College of Health Sciences, Mattu University, Mattu, Ethiopia,
2 Department of Lifelong Learning & Community Development, College of Education and Behavioral Science, Mattu University, Mattu, Ethiopia, 3 Department of Medical Laboratory, College of Health Sciences, Mattu University, Mattu, Ethiopia, 4 Department of Midwifery, College of Health Sciences, Mattu University, Mattu, Ethiopia, 5 Department of Nursing, College of Health Sciences, Mattu University, Mattu, Ethiopia, 6 Department of Computer Science, College of Engineering and Technology, Kebri Dehar University, Kebri Dahar, Ethiopia

* zemeneayenew@gmail.com

**Data Availability Statement:** All relevant data are within the manuscript and its Supporting Information files.

## Abstract

### Background

Digital literacy refers to the capacity to critically assess digital content, use digital tools in professional settings, and operate digital devices with proficiency. The healthcare sector has rapidly digitized in the last few decades. This systematic review and meta-analysis aimed to assess the digital literacy level of health professionals in the Ethiopian health sector and identify associated factors. The study reviewed relevant literature and analyzed the data to provide a comprehensive understanding of the current state of digital literacy among health professionals in Ethiopia.

### Methods

The study was examined by using the Preferred Reporting Items for Systematic Reviews and Meta-Analyses (PRISMA) criteria. Evidence was gathered from the databases of Google Scholar, Pub Med, Cochrane Library, Hinari, CINAHL, and Global Health. Consequently, five articles met the eligible criteria for inclusion. The analysis was carried out using STATA version 11. The heterogeneity was evaluated using the $I^2$ test, while the funnel plot and Egger's regression test statistic were used to examine for potential publication bias. The pooled effect size of each trial is evaluated using a random effect model meta-analysis, which provides a 95% confidence interval.

### Result

A total of five articles were included in this meta-analysis and the overall pooled prevalence of this study was 49.85% (95% CI: 37.22–62.47). six variables, Monthly incomes AOR =

**Funding:** The author(s) received no specific funding for this work.

**Competing interests:** There are no known competing financial interests or personal ties among the authors that could have influenced the work presented in this study.

3.89 (95% CI: 1.03–14.66), computer literacy 2.93 (95% CI: 1.27–6.74), perceived usefulness 1.68 (95% CI: 1.59–4.52), educational status 2.56 (95% CI: 1.59–4.13), attitude 2.23 (95% CI: 1.49–3.35), perceived ease of use 2.22 (95% CI: 1.52–3.23) were significantly associated with the outcome variable.

## Conclusion

The findings of the study revealed that the overall digital literacy level among health professionals in Ethiopia was relatively low. The study highlights the importance of addressing the digital literacy gap among health professionals in Ethiopia. It suggests the need for targeted interventions, such as increasing monthly incomes, giving computer training, creating a positive attitude, and educational initiatives, to enhance digital literacy skills among health professionals. By improving digital literacy, health professionals can effectively utilize digital technologies and contribute to the advancement of healthcare services in Ethiopia.

## Background

Digital technologies have revolutionized the healthcare industry, offering new opportunities for efficient and effective care delivery [1–3]. However, the successful integration and utilization of these technologies rely heavily on the digital literacy of healthcare professionals [4]. Digital literacy refers to the ability to use digital technologies proficiently, critically evaluate digital information, and apply digital tools in professional practice [4–6].

The healthcare sector has rapidly digitized in the last few decades and thanks to internet resources the ability to search, select, assess, and make use of digital applications and online health information is becoming more and more vital for healthcare consumers [4, 6, 7]. To research the implications of these competencies at the individual and population levels, a trustworthy digital health literacy assessment tool is needed [8]. When deciding how much a patient can benefit individually from particular eHealth tools and therapies, as in ordinary clinical practice, a measurement tool could be helpful [9, 10].

Digital literacy is a multifaceted skill set that empowers individuals to harness the full potential of digital technologies, tools, and resources [11]. In health sectors, Health professional's digital literacy encompasses the adept and efficient utilization of digital tools, technologies, and information systems within the healthcare context [12]. This proficiency involves a fusion of knowledge, skills, and attitudes necessary to navigate, evaluate, and leverage digital resources for diverse purposes [13]. From patient care and communication to data management and professional development, a digitally literate health professional possesses the expertise needed to maximize the benefits of technology in the dynamic landscape of healthcare [14]. A health professional's digital literacy level reflects their proficiency in using digital technologies within their practice [15]. It measures their overall understanding and capability to effectively utilize a range of digital tools and resources [11, 16]. A higher digital literacy level indicates advanced skills in navigating electronic health records, utilizing telemedicine platforms for remote consultations, and maintaining effective communication with patients through various digital channels [17].

In Ethiopia, as in many other developing countries, the healthcare sector is witnessing a rapid expansion of digital health initiatives [4, 18]. These initiatives aim to improve healthcare delivery, enhance data management, and strengthen health information systems [19, 20].

However, the digital literacy level of healthcare professionals in Ethiopian health sectors has not been thoroughly explored [21]. Studies done in Ethiopia indicated that there was a poor level of digital literacy among health professionals, and the outcomes ranged from 43.6% to 53.81% [11, 21–24]. This variation could be the health professional's computer literacy [21], monthly income [11, 23], educational status [22, 24], attitude [21], training access, awareness, willingness, perceived usefulness, perceived ease of use and others [11, 21–24]. A prominent challenge in Ethiopia is the restricted access to digital infrastructure and resources within healthcare facilities [25]. Particularly in rural areas, numerous health professionals lack reliable internet connectivity and essential digital devices [26]. This limited access significantly impedes their capacity to develop and improve their digital literacy level [27].

However, there are ongoing efforts to improve digital literacy among health professionals in Ethiopia [27]. The Ministry of Health has recognized the importance of digital technologies in healthcare and has initiated various programs to enhance digital literacy [18]. For instance, the Ethiopian Health Informatics Association (EHIA) has been established to promote the use of digital technologies in healthcare and provide training and support to health professionals [20].

Moreover, it's crucial to grasp the digital literacy levels and factors influencing healthcare professionals in Ethiopia. This understanding is key for effectively implementing digital health technologies [9, 10]. It enables us to spot knowledge and skill gaps, create targeted training, and shape policies. This ensures healthcare professionals are ready to use digital tools for better patient care [28, 29].

This systematic review and meta-analysis aim to gauge the digital literacy of Ethiopian healthcare professionals and explore factors influencing their levels [30]. Through synthesizing existing evidence, this study aims to give a comprehensive view of digital literacy among Ethiopian healthcare professionals.

These findings are significant for healthcare policy, education, and practice. The study not only identifies gaps but also informs policy decisions, contributing to a more informed and digitally adept healthcare workforce. Conducting this study in Ethiopia is crucial for improving healthcare delivery, and decision-making, and adding to our knowledge base. Ultimately, it paves the way for enhanced digital literacy among healthcare professionals, benefiting both providers and patients in Ethiopia.

## Methods and materials

### Source of information and search strategy

The Preferred Reporting Items for Systematic Reviews and Meta-Analysis (PRISMA) checklist was used to conduct the pooled prevalence and associated factors toward digital health literacy and its associated factors in Ethiopia [31]. The study team developed a review strategy between May 2023 and July 2023 and a database search was done online. We searched pertinent published publications using the databases; MEDLINE, PubMed, Embrace, Web of Science, Scopus, and Cochrane Library. A systematic evaluation based on the following combinations of search terms was conducted to search publications in online databases: ("digital health literacy" OR "eHealth literacy" OR "electronic health literacy" OR "digital literacy" OR "digital literacy level") AND ("Associated factors").

### Eligibility criteria

The study analysis includes original research studies that reported on health professionals' digital literacy level and associated factors in Ethiopia. Furthermore, studies done by a quantitative and mixed quantitative supported by qualitative study methodology on health professionals in all healthcare settings were included. Research published in peer-reviewed

journals or published between December 2015 to July 2023 and the English language is also mentioned. Literature primarily focuses on health professionals' digital literacy level and identifies factors related to the outcome variables were included. The reason for excluding researchers without editorial reports, letters, reviews, or commentaries from the study is because these types of publications generally do not contain original research data or provide significant insights into the digital literacy levels of health professionals in Ethiopian health sectors. This exclusion helps maintain the focus of the research on primary studies that directly contribute relevant data to the systematic review and meta-analysis. By omitting such publications, the study can ensure that only robust evidence is included, thereby enhancing the reliability and validity of the findings. Additionally, excluding literature with incomplete texts, difficult-to-extract data, out-of-English-language publications, and uncategorized outcome variables helps streamline the research process and ensure that only high-quality, relevant information is analyzed.

## Data extraction

A common computer-based spreadsheet including the data from the included studies was created by one investigator and a second researcher to check the consistency. The initial author's name, the year the study was published, the number of participants, details about the participants' backgrounds, study areas, the sample size, data collection methods, and the study's design were all collected for each study. With 95% confidence intervals, the magnitude of the effect of short message service (SMS) on digital health literacy and its associated factors in Ethiopia and related parameters were also extracted and any difficulties encountered while extracting the data will be discussed with the corresponding author.

## Evaluation of the selected literature's quality

Using a standardized tool (a modified version of the Newcastle-Ottawa Scale (NOS)), that categorizes bias potential and can help to explain variations in the results of included research, Each study's quality was assessed, and the authors additionally examined at each publication's methodology and other features [32]. We concluded that works with a modified NOS component score of 7 or higher were relevant after analyzing a range of publications within nine quality assessment criteria questions (See S1 Table) [33]. Additionally, three authors independently conducted a quality control assessment.

## Data processing and analysis

The information that was gathered was initially exported from Microsoft Excel and imported into STATA version 11. To perform further analysis we used a random-effect model of meta-analysis that estimates the pooled effect size and effect of each study along with their corresponding 95% confidence intervals (CI). To visualize the data, forest plots were utilized to assess the pooled impact size and weight with 95% CI of each selected study. The degree of heterogeneity between the included studies was evaluated using the indicator of heterogeneity (I2 statistics) [34]. We also employed a random-effect model to accommodate for significant heterogeneity of assessed the included studies in terms of participants, locations, and measurement. To check for publication bias in the meta-analysis using funnel plot and Egger's test [33].

### Ethical approval

Ethics clearance was not mandatory for systematic review and meta-analysis due to the study's nature. As we focused on analyzing existing data without direct involvement with human subjects, no interventions, interactions, or identifiable private information were parts of our research. Consequently, the study was exempt from ethics clearance, aligning with ethical standards. This decision underscores our commitment to the responsible conduct of research, prioritizing the protection of participants' rights and well-being.

## Result

### The selection process of the articles

Searching Google, Google Scholar, and other online search engines yielded a total of 6,312 papers across all databases (Medline, Pub Med, Scopus, Cochrane, EMBASE, African Journal Online (AJOL), HINARI, and Science Direct). Due to duplication, 1251 of them were removed. In addition, 3775 papers were disqualified following an examination of their abstracts and titles. Furthermore, 78 publications were excluded due to the paper's quality and lack of full-text availability, while 1,202 full-text papers were excluded due to the study area. Eventually, a meta-analysis and systematic review were conducted on five (5) full-text papers (See: Fig 1).

### Characteristics of the included articles in the review

A total of 5 articles with 1,938 study participants were included in this study to estimate the pooled level of digital literacy among healthcare professionals in Ethiopian health sectors. Most of the studies included in this systematic review and meta-analysis were done in the Amhara region with study participants varying from 193 to 476 [11, 21, 23]. The rest two articles were done in Addis Ababa and the Oromia region, with total study participants of 846 [22, 24]. All of those included studies were conducted among health professionals and used an institutional-based cross-sectional study design. The magnitude of health professionals' digital literacy level varied from 43.6% to 53.81% with the included articles in Ethiopia. Furthermore, all reviewed articles meet the specified quality, which was seven and above, according to the Joanna Briggs Institute quality score assessment (See Table 1).

### The pooled prevalence of Ethiopian health professionals' high digital literacy level

According to the reviewed articles, health professionals' digital literacy level in the Ethiopian health sector was low. Based on this meta-data analysis, the Ethiopian health professionals' high digital literacy level pooled prevalence was 49.85% (95% CI: 37.22–62.47). A statistically significant amount of heterogeneity was not identified using a random-effects model ($I2 = 0.00\%$; $p = 0.99$). As a result, this data demonstrated that there was no significant heterogeneity across the primary studies and that subgroup analysis was not necessary (See Fig 2).

### Publication bias

Egger's regression test and funnel plot were used to determine whether a publishing bias was present or absent. Funnel plots in this meta-analysis showed evidence of publication bias. A symmetrical distribution indicates that there is no publishing bias, and a visual examination of the funnel plot further shows a symmetry distribution. Each point in a funnel plot does not

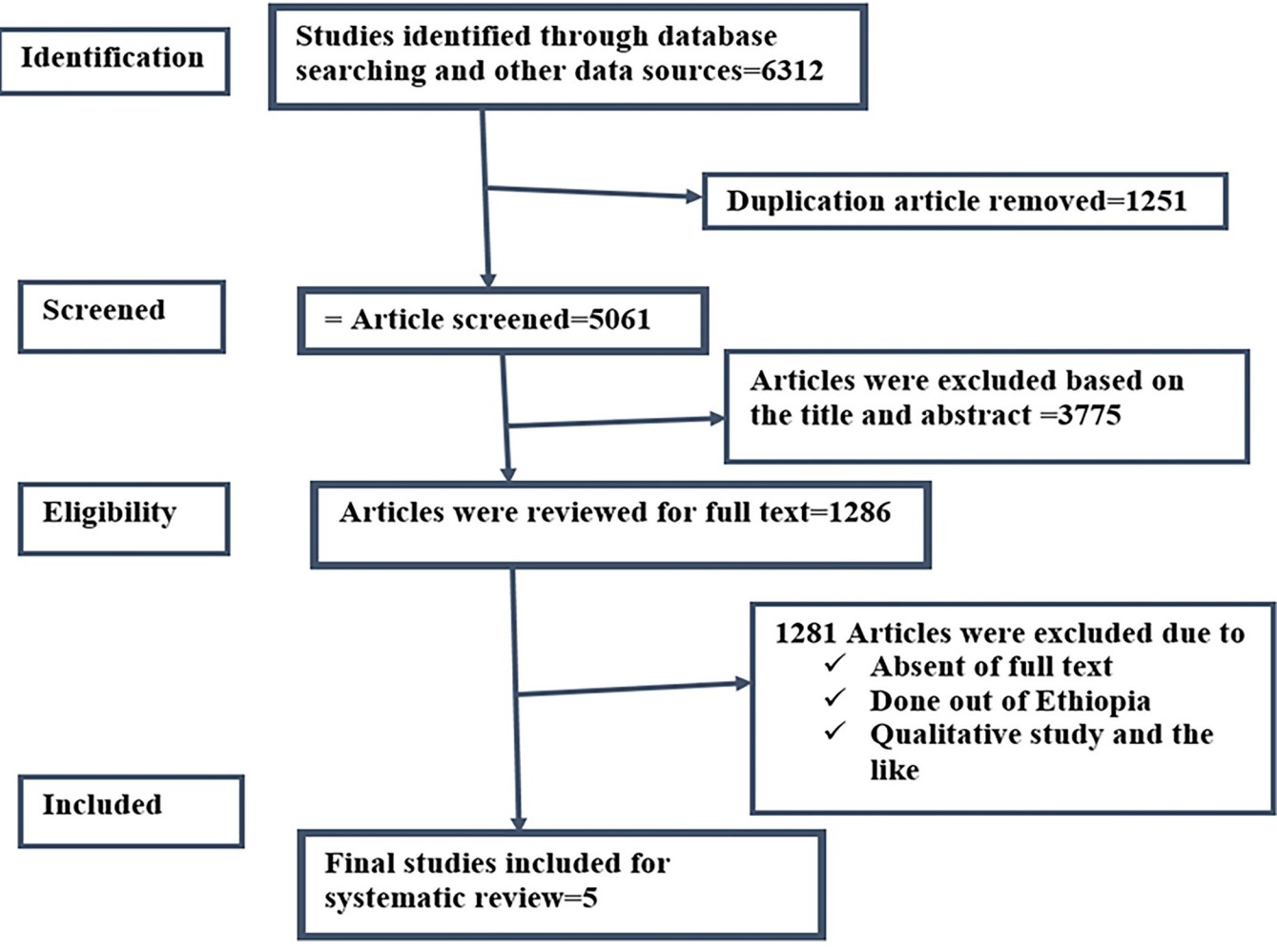

**Fig 1. PRISMA flowcharts showing the selection process of the articles.**

represent a separate study. Similarly, Egger's test result (P = 0.022) for the absence of publication bias was not statistically significant. (See Fig 3).

## Factors associated with Ethiopian health professionals' digital literacy level

This study determined several factors that related to Ethiopian health professionals' high digital literacy level within five (5) articles. Accordingly, six associated factors were identified. Monthly incomes of Ethiopian health professionals with Adjusted odd ratio (AOR) = 3.89 (95% CI: 1.03–14.66), computer literacy of the respondents AOR = 2.93 (95% CI: 1.27–6.74),

**Table 1. Characteristics of individual studies conducted on digital literacy level among health professionals in Ethiopia, 2023.**

| Authors | Regions | Year of the study | Year of Publication | Study design | sample size | Prevalence | Quality score |
|---|---|---|---|---|---|---|---|
| Ahmed. M et al | Oromia | 2021 | 2022 | cross-sectional | 423 | 43.6 | 9 |
| Shiferaw. K et al | Amhara | 2019 | 2020 | cross-sectional | 193 | 49.7 | 8 |
| Tegegne. M et al. | Amhara | 2022 | 2023 | cross-sectional | 423 | 51.8 | 9 |
| Chereka. A et al | Amhara | 2021 | 2022 | cross-sectional | 476 | 50.4 | 8 |
| Assaye. B et al. | Addis Ababa | 2020 | 2023 | cross-sectional | 423 | 53.81 | 7 |

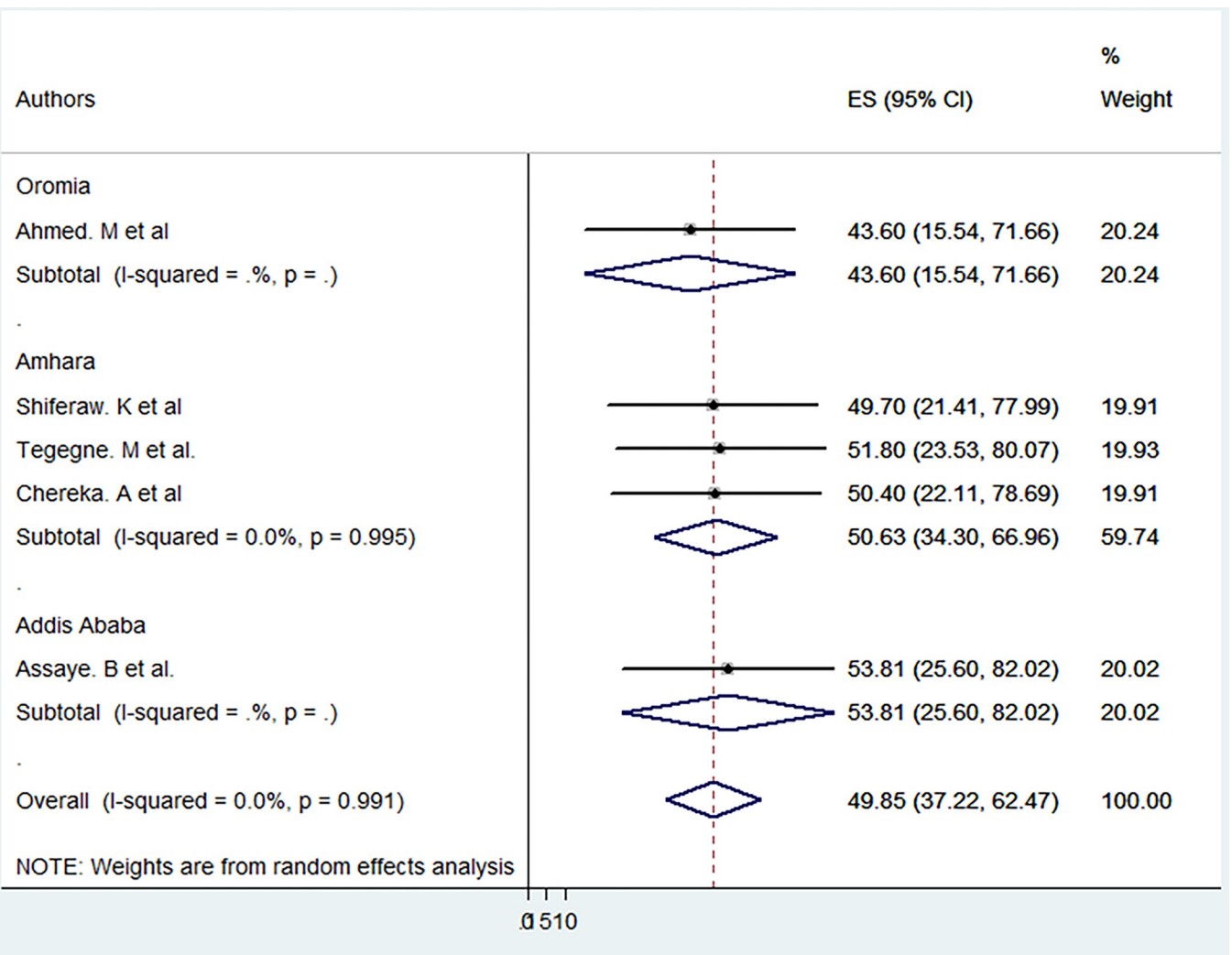

**Fig 2. The pooled prevalence of high digital literacy levels among health professionals in Ethiopia.** 2023.

perceived usefulness AOR = 1.68 (95% CI: 1.59–4.52), educational status AOR = 2.56 (95% CI: 1.59–4.13), attitude AOR = 2.23 (95% CI: 1.49–3.35), perceived ease of use AOR = 2.22 (95% CI: 1.52–3.23) were factors associated with health professional's digital literacy level among the selected articles.

Four studies in total were accessed to evaluate the relationship between health professionals' digital literacy level and educational status, three studies were conducted to examine the relationship between favorable attitude and health professionals' digital literacy level, and two studies were conducted to evaluate computer literacy level, perceived usefulness, perceived ease of use and monthly incomes of Ethiopian health professionals and digital literacy level (See Fig 4).

## Discussion

This study assessed the Ethiopian health professionals' digital literacy levels through a systematic review and meta-analysis. Accordingly, this finding revealed that the overall health professionals' higher digital literacy level in the Ethiopian health sector was 49.85% (95% CI: 37.22–

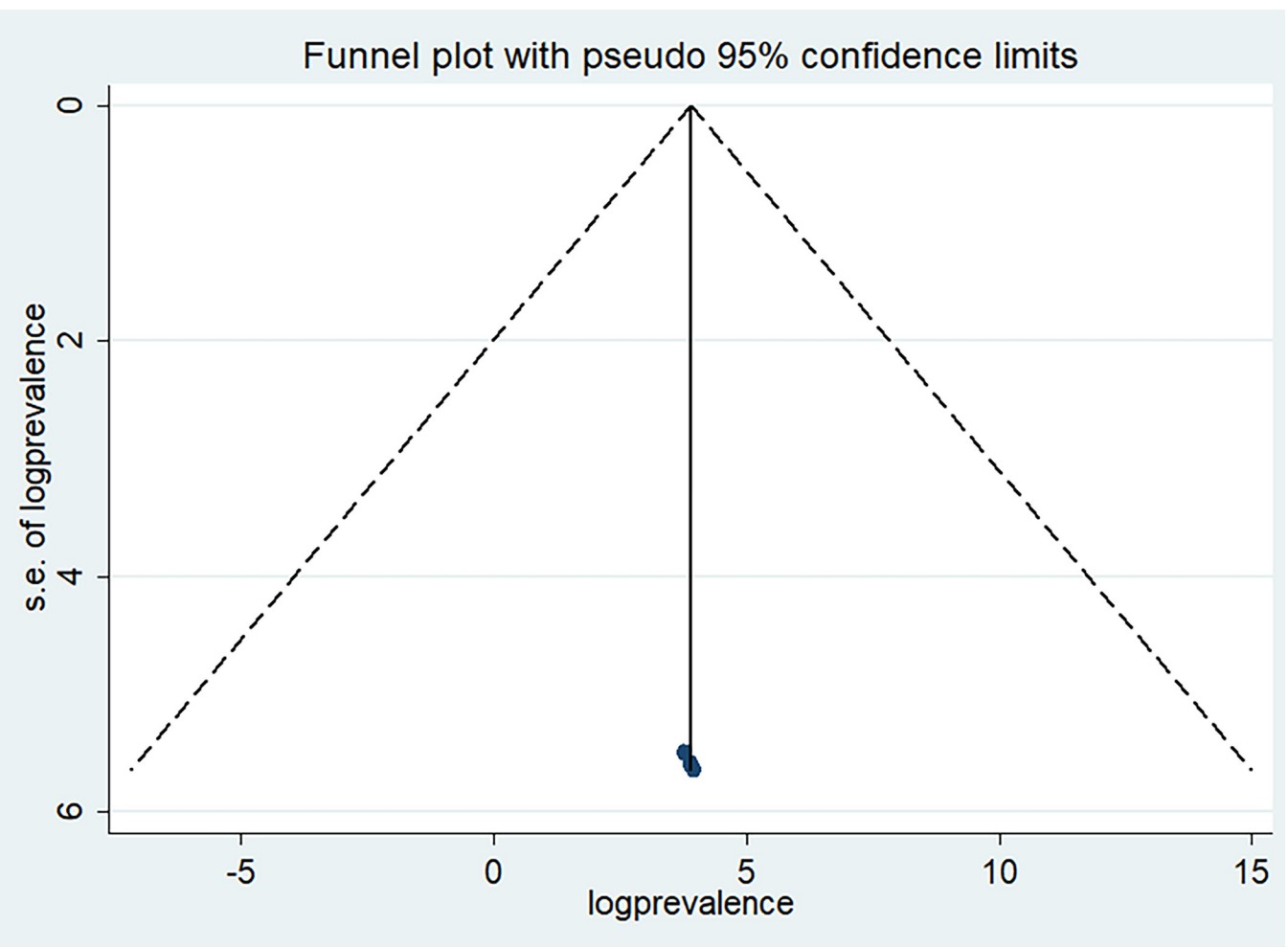

**Fig 3. Funnel plot to shows publication bias among the included articles of high digital literacy level a systematic review and meta-analysis, Ethiopia, 2023.**

62.47). These findings were derived from meticulous literature searches and analysis, ensuring a comprehensive evaluation and quantitative summary [35]. Such insights offer valuable evidence for interventions and assist decision-makers in effectively addressing digital literacy requirements within the healthcare sector [36–38].

Based on the associated factors analysis, six factors were significantly associated with health professionals' digital literacy levels in Ethiopia. Among those, the monthly incomes of the respondents were substantially significantly associated with health professionals' digital literacy level in Ethiopia. Respondents who had higher monthly incomes were 3.89 times more digitally literate than the respondents who had lower monthly incomes. This could be attributed to the fact that individuals with higher incomes may have better access to technology and resources, which allows them to develop and enhance their digital skills. It was in line with the study done [22, 23].

The second most significant associated factor with Ethiopian health professional's digital literacy level was computer literacy. Respondents who were computer literate were 2.93 times more digital literate than the respondents who were low computer literate. This could be because health professionals who possess better computer literacy skills are more likely to have

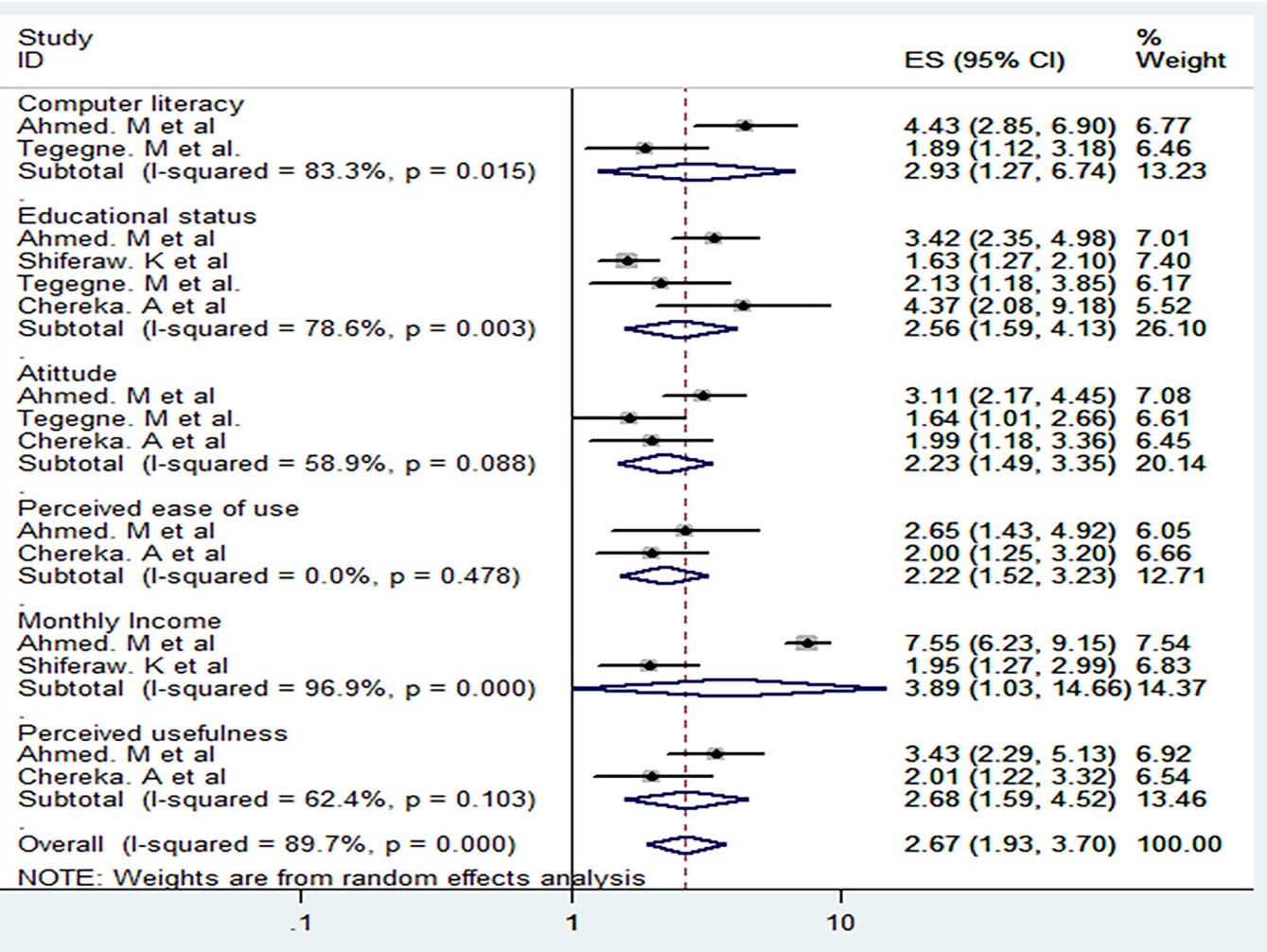

**Fig 4. Factors associated with Ethiopian health professionals' digital literacy level.**

higher digital literacy levels. Computer literacy refers to the ability to use computers and related technologies effectively. It was in line with the study done [11, 22].

Respondents who perceived digital tools as useful were 2.68 times higher in digital literacy level than that of respondents who perceived digital tools were not useful. This could be because Health professionals who perceive digital tools as valuable and beneficial to their work are more likely to invest time and effort in developing their digital literacy. This was consistent with the study done [21, 22].

Respondents who were masters and above holders had 2.56 times higher digital literacy than respondents who were BSc and below holders, the study suggests that health professionals with higher educational levels tend to have better digital literacy skills. Higher education provides individuals with a foundation for acquiring and utilizing digital skills. Those highly educated health professionals use digital technology for their educational purposes; such includes research and data collection tools. This was in line with the study done [11, 21–23].

Health professionals with a favorable attitude toward digital health technology were 2.23 times more likely to have high digital literacy than those with unfavorable attitudes. The attitude of health professionals towards digital technology plays a significant role in their digital

literacy level. Positive attitudes towards technology are associated with higher digital literacy, as individuals are more motivated to learn and adapt to new digital tools and platforms. This was supported by a study done [11, 21, 22].

The last significant associated factor with this in the current study was perceived ease of use. Respondents who perceived using digital health tools as being easy were 2.22 times more likely to have a higher digital health literacy level than their counterparts. This could be because health professionals who perceive digital technology as easy to use are more likely to have higher digital literacy levels. When individuals find technology user-friendly and intuitive, they are more inclined to engage with it and develop their digital skills. It was in line with the study done [21, 22].

## Conclusion

This systematic review and meta-analysis highlight the significance of various factors in determining health professionals' digital literacy levels in Ethiopian health sectors. The study revealed that the overall digital literacy level among health professionals in Ethiopia was 49.85%. Which was relatively low, Monthly income, computer literacy, educational status, attitude, perceived ease of use, and perceived usefulness were identified as significant factors influencing digital literacy levels. Understanding these factors can help inform strategies and interventions aimed at improving digital literacy among health professionals in Ethiopia.

Policymakers, healthcare institutions, and relevant stakeholders need to recognize the importance of digital literacy and prioritize initiatives that address the identified factors. By doing so, Ethiopia can empower its health professionals to navigate the digital landscape and leverage technology to its full potential, ultimately contributing to the advancement of the healthcare sector in the country. Further research and continuous monitoring of digital literacy levels are necessary to track progress and inform future interventions.

## Strength and weakness

The current study provides a comprehensive assessment of the digital literacy level among health professionals in Ethiopian health sectors. It followed a systematic approach, including a thorough search strategy, selection criteria, and quality assessment of included studies. This enhances the reliability and validity of the findings. Additionally, it allows for the quantitative synthesis of data from multiple studies, providing a more robust estimation of the overall effect size and identifying significant factors associated with digital literacy.

As a limitation, there were a limited number of studies included in this study. It may affect the overall reliability and generalizability of the findings and the review's strength heavily relies on the availability and quality of the included studies. The review may be susceptible to publication bias, as studies with significant results are more likely to be published, while studies with non-significant or negative results may remain unpublished. This bias can influence the overall effect size and the interpretation of the findings. Furthermore, the findings of this review may be specific to the Ethiopian health sectors and may not be directly applicable to other countries or regions with different healthcare systems, resources, and digital literacy initiatives.

## Supporting information

**S1 Table. Quality assessment of the selected studies.**
(DOCX)

**S2 Table. PRISMA check list.**
(DOCX)

**S1 Dataset. Prevalence dataset.**
(XLSX)

**S2 Dataset. Factors associated dataset.**
(XLSX)

## Acknowledgments

We express our gratitude to each author of the papers that were part of our meta-analysis and systematic review.

## Author Contributions

**Conceptualization:** Alex Ayenew Chereka, Sisay Yitayih Kassie, Adamu Ambachew Shibabaw, Teshome Bekana, Gemeda Wakgari Kitil.

**Data curation:** Alex Ayenew Chereka, Agmasie Damtew Walle, Sisay Yitayih Kassie, Adamu Ambachew Shibabaw, Addisalem Workie Demsash, Teshome Bekana, Gemeda Wakgari Kitil, Mathias Nega Tadesse.

**Formal analysis:** Alex Ayenew Chereka, Agmasie Damtew Walle, Sisay Yitayih Kassie, Adamu Ambachew Shibabaw, Addisalem Workie Demsash, Teshome Bekana, Gemeda Wakgari Kitil, Mathias Nega Tadesse.

**Funding acquisition:** Alex Ayenew Chereka.

**Investigation:** Alex Ayenew Chereka, Gemeda Wakgari Kitil.

**Methodology:** Alex Ayenew Chereka, Gemeda Wakgari Kitil.

**Software:** Alex Ayenew Chereka, Agmasie Damtew Walle, Sisay Yitayih Kassie, Mathias Nega Tadesse.

**Supervision:** Alex Ayenew Chereka, Gemeda Wakgari Kitil.

**Validation:** Alex Ayenew Chereka, Sisay Yitayih Kassie, Abiy Tassew Dubale, Teshome Bekana, Gemeda Wakgari Kitil, Milkias Dugassa Emanu.

**Visualization:** Alex Ayenew Chereka, Agmasie Damtew Walle, Addisalem Workie Demsash, Mekonnen Kenate Hunde, Teshome Bekana, Milkias Dugassa Emanu.

**Writing – original draft:** Alex Ayenew Chereka, Fikadu Wake Butta, Mekonnen Kenate Hunde, Abiy Tassew Dubale, Gemeda Wakgari Kitil, Milkias Dugassa Emanu.

**Writing – review & editing:** Alex Ayenew Chereka, Fikadu Wake Butta, Addisalem Workie Demsash, Mekonnen Kenate Hunde, Abiy Tassew Dubale, Teshome Bekana, Milkias Dugassa Emanu, Mathias Nega Tadesse.

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
