## [Decision Letter · Decision Letter 0]

27 Dec 2023

PONE-D-23-38333Dear PLOS ONE; 

Assessing health professional’s digital literacy level and associated factors in Ethiopian health sectors: A systematic review and Meta-analysisPLOS ONE

Dear Dr. Chereka,

Thank you for submitting your manuscript to PLOS ONE. After careful consideration, we feel that it has merit but does not fully meet PLOS ONE’s publication criteria as it currently stands. Therefore, we invite you to submit a revised version of the manuscript that addresses the points raised during the review process.

We look forward to receiving your revised manuscript.

Kind regards,

Jahanpour Alipour, Ph.D.

Academic Editor

PLOS ONE

2. In the online submission form, you indicated that [All of the data used and analyzed for this work are available from the corresponding author upon reasonable request.]. 

Reviewers' comments:

Reviewer's Responses to Questions

**Comments to the Author**

1. Is the manuscript technically sound, and do the data support the conclusions?

Reviewer #1: Yes

Reviewer #2: Partly

2. Has the statistical analysis been performed appropriately and rigorously? 

Reviewer #1: Yes

Reviewer #2: Yes

3. Have the authors made all data underlying the findings in their manuscript fully available?

Reviewer #1: No

Reviewer #2: Yes

4. Is the manuscript presented in an intelligible fashion and written in standard English?

Reviewer #1: Yes

Reviewer #2: Yes

5. Review Comments to the Author

Reviewer #1: 1. The importance of the topic is not clearly stated. The author should further describe the current situation of digital literacy of local medical staff and the significance of the topic, with emphasis on the innovation of the topic.

2. The inclusion and exclusion criteria are not clear enough and need to be described in detail. The author should make a more specific elaboration from the types of studies, research methods and outcome indicators.

3. The number of selected papers is relatively small, which will have a certain impact on the accuracy of the results, and the author should explain the corresponding reasons and shortcomings.

Reviewer #2: The manuscript examines a very important topic and its results can be generalized to communities with limited resources. The main weakness of this study is the non-transparent search process and databases examined. It is also well written and easy to understand for me as a non-native person. However, there are a number of minor errors, the resolution of which will lead to a higher quality of the article.

Background

It is suggested to delete the first paragraph of the background as it is not related to the purpose of the study and contains repetitive and old material.

In the third paragraph, the text discusses the digital literacy assessment tool. It is unclear why this tool is being discussed and whether its development was one of the items of interest in the study. It would be more effective to focus on digital literacy and its components.

The last sentences of the last paragraph pertain to the conclusion section.

Methods and material

Explain how to score according to NOS and the number of questions.

Results

The databases reported in this section differ from those mentioned in the method.

The search process appears to have been conducted incorrectly. If Ethiopia is one of the key search terms, as mentioned in the method, studies outside of this country should not have been included in the initial search. It is unclear why studies outside of Ethiopia were excluded in the eligibility stage.

To ensure clarity, it is recommended to present the dimensions of digital literacy mentioned in each study in a table format.

Discussion

The study's results should be compared to those of similar studies, and conclusions should be drawn based on the comparison. It is not advisable to repeat sentence A.

6. PLOS authors have the option to publish the peer review history of their article (what does this mean?). If published, this will include your full peer review and any attached files.

Reviewer #1: No

Reviewer #2: **Yes: **Mohammad Hosein Hayavi-haghighi

---

## [Author Response · Author response to Decision Letter 0]

16 Jan 2024

Dear Editors of PLOS ONE;

 It has been recalled that we the authors of the manuscript entitled “Assessing Health Professional’s Digital Literacy Level and Associated Factors in Ethiopian Health Sectors: A Systematic Review and Meta-analysis” submitted our manuscript for publication in your journal received reviewer comments for the betterment of the manuscript before its publication. 

In line with this, all authors are very happy with the constructive and valuable comments given by reviewers. Accordingly, we have considered all the comments and provided a point-by-point response and explanations for all the questions raised. 

Finally, we have submitted all the required documents in their revised form. We hope that we have addressed all the questions and if you have any points for further clarity, let us know. 

All the authors would like to thank you the editorial team and the reviewers.

Editor(s)' Comments to Authors

Answer for reviewer 1

Comment 1: The importance of the topic is not clearly stated. The author should further describe the current situation of digital literacy of local medical staff and the significance of the topic, with emphasis on the innovation of the topic.

Answer: Thank you very much for your interesting and supportive comments; we have explained the importance of the topic in detail (pages 3-5).

Comment 2: The inclusion and exclusion criteria are not clear enough and need to be described in detail. The author should make a more specific elaboration on the types of studies, research methods, and outcome indicators.

Answer: Thank you so much for your constructive comments. We have tried to clarify the comments that you raised to us (page 6, lines 12- 21).

Comment 3: The number of selected papers is relatively small, which will have a certain impact on the accuracy of the results, and the author should explain the corresponding reasons and shortcomings.

Answer: Thank you for your valuable feedback regarding the number of articles included in our assessment of digital literacy levels. We appreciate your concern regarding the relatively small number of selected papers in our study. However, our extensive search across various databases, browsers, and search engines, we found limited research conducted on the digital literacy level in Ethiopia. This scarcity of available studies is a reflection of the current state of research in this specific context. We acknowledge this limitation and will provide a clear explanation of the reasons and shortcomings in our manuscript. Due to this, we conclude the number of research related to digital literacy levels is limited in Ethiopia. Thank you for bringing this to our attention.

Answer for reviewer 2

Comment 1: It is suggested to delete the first paragraph of the background as it is not related to the purpose of the study and contains repetitive and old material.

Answer: Thank you for your valuable feedback. We have removed the recommended paragraph and briefly explained it in other ways (page 3).

Comment 2: In the third paragraph, the text discusses the digital literacy assessment tool. It is unclear why this tool is being discussed and whether its development was one of the items of interest in the study. It would be more effective to focus on digital literacy and its components.

Answer: Thank you so much for your supportive comments. We have modified the ideas of the sentence based on the given comment and explained more about digital literacy and digital literacy level (pages 3-4).

Comment 3: The last sentences of the last paragraph pertain to the conclusion section.

Answer: Thank you for your valuable feedback and detailed comments: we have modified the sentences you suggested to remove (page 5) 

Comment 4: Explain how to score according to NOS and the number of questions.

Answer: Thank you so much for your interesting comment. We have clearly explained and attached the quality assessment criteria questions as supportive material (page 7).

Comment 5: The databases reported in this section differ from those mentioned in the method.

The search process appears to have been conducted incorrectly. If Ethiopia is one of the key search terms, as mentioned in the method, studies outside of this country should not have been included in the initial search. It is unclear why studies outside of Ethiopia were excluded in the eligibility stage. To ensure clarity, it is recommended to present the dimensions of digital literacy mentioned in each study in a table format.

Answer: Thank you very much for your advisable comment. We have corrected this in the method section. I.e. we have removed the keyword Ethiopia from the method section and modified other keywords (page 6).

Comment 6: The study's results should be compared to those of similar studies, and conclusions should be drawn based on the comparison. It is not advisable to repeat sentence A.

Answer: Thank you for your valuable feedback. We appreciate your suggestion to compare our study's results to those of similar studies and draw conclusions based on the comparison. However, despite our comprehensive search efforts, we were unable to find any systematic review papers specifically related to digital literacy level. We acknowledge the limitation and understand the importance of comparative analysis. We will explore alternative approaches to effectively compare results and draw meaningful conclusions. Thank you for your guidance in this matter.

---

## [Decision Letter · Decision Letter 1]

19 Feb 2024

PONE-D-23-38333R1Dear PLOS ONE; 

Assessing health professional’s digital literacy level and associated factors in Ethiopian health sectors: A systematic review and Meta-analysisPLOS ONE

Dear Dr. Chereka,

Thank you for submitting your manuscript to PLOS ONE. After careful consideration, we feel that it has merit but does not fully meet PLOS ONE’s publication criteria as it currently stands. Therefore, we invite you to submit a revised version of the manuscript that addresses the points raised during the review process.

We look forward to receiving your revised manuscript.

Kind regards,

Jahanpour Alipour, Ph.D.

Academic Editor

PLOS ONE

Journal Requirements:

Reviewers' comments:

Reviewer's Responses to Questions

**Comments to the Author**

1. If the authors have adequately addressed your comments raised in a previous round of review and you feel that this manuscript is now acceptable for publication, you may indicate that here to bypass the “Comments to the Author” section, enter your conflict of interest statement in the “Confidential to Editor” section, and submit your "Accept" recommendation.

Reviewer #2: All comments have been addressed

Reviewer #3: (No Response)

2. Is the manuscript technically sound, and do the data support the conclusions?

Reviewer #2: Yes

Reviewer #3: (No Response)

3. Has the statistical analysis been performed appropriately and rigorously? 

Reviewer #2: Yes

Reviewer #3: (No Response)

4. Have the authors made all data underlying the findings in their manuscript fully available?

Reviewer #2: Yes

Reviewer #3: (No Response)

5. Is the manuscript presented in an intelligible fashion and written in standard English?

Reviewer #2: Yes

Reviewer #3: (No Response)

6. Review Comments to the Author

Reviewer #2: The authors have made significant edits, but the discussion section requires further improvement. It should include more references and employ more reasoning and deduction. For example, a sample from the discussion section of the reference below could be used. Best of luck.

Hayavi-Haghighi MH, Alipour J. Applications, opportunities, and challenges in using Telehealth for burn injury management: A systematic review. Burns. 2023 Sep;49(6):1237-1248. doi: 10.1016/j.burns.2023.07.001. Epub 2023 Jul 13. PMID: 37537108.

Reviewer #3: Dear Author

Thank you to the authors for choosing a current and relevant topic for the manuscript related to health issues. The following suggestions are proposed to improve the quality of the manuscript.

- It's better to seek help from a native English speaker to edit the text.

- The short title is not short enough; it's better to make it shorter.

- In the background section, remove the letter S at the beginning of the first line.

- In the background section, the reference numbers should start from 1, but they have started from 11. Please review and edit all the references in manuscript.

- In the last line of the "eligibility criteria" section, you wrote "Additionally, researchers without editorial reports, letters, reviews, or commentaries were excluded from the study. Why? Please explain about the reason.

- In the "eligibility criteria" section, it is stated that studies from 2015 onwards have been considered. Please explain why studies before 2015 were not considered.

- The title of a table 1 be written as: Quality evaluation assessment of the selected study. It is better to remove one of the words; “assessment or evaluation”. Evaluation assessment is not correct. also replace the “study” with “studies”.

- Please modify the title of S1 Table 1. Using the words " assessment " and " evaluation " consecutively is not correct.

7. PLOS authors have the option to publish the peer review history of their article (what does this mean?). If published, this will include your full peer review and any attached files.

Reviewer #2: No

Reviewer #3: No

---

## [Author Response · Author response to Decision Letter 1]

20 Feb 2024

Dear Reviewers

On behalf of the authors, I extend our sincere gratitude for your meticulous review of our manuscript, "Evaluating Digital Literacy of Health Professionals in Ethiopian Health Sectors: A Systematic Review and Meta-Analysis." Your expertise and insightful feedback have played an instrumental role in enhancing the clarity and quality of our work. Your commitment to excellence and professionalism throughout the review process has not gone unnoticed, and we are deeply appreciative of your dedication. Your willingness to share your expertise has been invaluable to us, and we recognize the significant contribution you have made to our research. Your thorough examination and constructive critique have undoubtedly strengthened the manuscript.

Once again, we express our heartfelt thanks for your invaluable contribution to our work. Your input has been crucial in shaping the final outcome, and we are genuinely grateful for your commitment to advancing scientific knowledge.

Reviewers' Comments to Authors

Answer for reviewers comments 

Reviewer 2: The authors have made significant edits, but the discussion section requires further improvement. It should include more references and employ more reasoning and deduction.

Answer: Thank you for your valuable feedback and detailed comments regarding discussion section. We have tried to improve based on the given comments (Page 9 line 17 to 20).

Reviewer 3:

Comment 1: - It's better to seek help from a native English speaker to edit the text. 

Answer: Thank you for your valuable feedback on our manuscript, your insights have been incredibly helpful in improving the clarity and quality of our work. We have taken your suggestions seriously, particularly regarding the need for language editing to ensure clear and accurate communication. 

To address this, we are currently seeking assistance from a native English speaker to edit the text thoroughly. We appreciate your patience and understanding as we strive to enhance the readability and coherence of the manuscript. Your continued support is invaluable to us, and we are committed to addressing all concerns raised during the review process.

Comment 2: The short title is not short enough; it's better to make it shorter.

Answer: Thank you for your valuable feedback regarding the short title of our manuscript. We understand the importance of brevity and have made further adjustments to ensure the title is concise and accurately reflects the content of our research. We appreciate your guidance and attention to detail throughout the editorial process.

Comment 3: In the background section, remove the letter S at the beginning of the first line.

Answer: Thank you for your valuable feedback to removing unnecessary character in our manuscript. We remove the later s in the background section (page 3 line 2)

Comment 4: In the background section, the reference numbers should start from 1, but they have started from 11. Please review and edit all the references in manuscript.

Answer: Thank you for bringing to our attention the issue with the reference numbers in the background section of our manuscript. We have carefully reviewed and corrected all references to ensure they start from 1 rather than 11. Additionally, we have thoroughly updated the entire manuscript to reflect these changes. We appreciate your diligence in ensuring the accuracy of our work.

Comment 5: In the last line of the "eligibility criteria" section, you wrote "Additionally, researchers without editorial reports, letters, reviews, or commentaries were excluded from the study. Why? Please explain about the reason.

Answer: Thank you for your insightful feedback regarding the "eligibility criteria." Excluding papers such as non-editorial reports, letters, reviews, or commentaries from the study serves to uphold the integrity and focus of our research. These types of publications typically lack original research data or offer substantive insights into the digital literacy levels of health professionals in Ethiopian health sectors. By omitting them, our study ensures that only primary research studies containing relevant data are included, thereby enhancing the reliability and validity of the systematic review and meta-analysis. This focused approach enables us to gather robust evidence directly contributing to our understanding of digital literacy among health professionals in Ethiopia, rather than including potentially tangential or less relevant information (page 5 lines 21 to page 6 lines 2).

Comment 6: In the "eligibility criteria" section, it is stated that studies from 2015 onwards have been considered. Please explain why studies before 2015 were not considered.

Answer: Thank you for your insightful feedback regarding the "eligibility criteria." In Ethiopia, the landscape of healthcare and technology has undergone significant transformations in recent years. Before 2015, there were limited initiatives and investments in digital health infrastructure and policies. Consequently, studies conducted before this period may not accurately reflect the current state of digital literacy among health professionals or the effectiveness of related interventions. By focusing on studies from 2015 onwards, we ensure relevance to the contemporary Ethiopian healthcare context, where digital literacy initiatives have gained momentum and are more representative of the current situation. This approach allows us to provide up-to-date and pertinent insights for informing future policies and interventions in the Ethiopian health sector.

Comment 7: - The title of a table 1 is written as: Quality evaluation assessment of the selected study. It is better to remove one of the words; “assessment or evaluation”. Evaluation assessment is not correct. Also replace the “study” with “studies”.

Answer: Thank you for your valuable feedback to removing unnecessary words in the table caption. We have removed and correct it based on your suggestion (page 6 line 11)

Comment 8: Please modify the title of S1 Table 1. Using the words “assessment “and” evaluation " consecutively is not correct.

Answer: Thank you for your valuable feedback to removing unnecessary words in the table caption. We have removed and correct it based on your suggestion (page 11 line 27)

---

## [Decision Letter · Decision Letter 2]

27 Feb 2024

Dear PLOS ONE;  Evaluating Digital Literacy of Health Professionals in Ethiopian Health Sectors: A Systematic Review and Meta-Analysis

PONE-D-23-38333R2

Dear Alex Ayenew Chereka,

We’re pleased to inform you that your manuscript has been judged scientifically suitable for publication and will be formally accepted for publication once it meets all outstanding technical requirements.

Kind regards,

Jahanpour Alipour, Ph.D.

Academic Editor

PLOS ONE

**Comments to the Author**

1. If the authors have adequately addressed your comments raised in a previous round of review and you feel that this manuscript is now acceptable for publication, you may indicate that here to bypass the “Comments to the Author” section, enter your conflict of interest statement in the “Confidential to Editor” section, and submit your "Accept" recommendation.

Reviewer #2: All comments have been addressed

Reviewer #3: (No Response)

2. Is the manuscript technically sound, and do the data support the conclusions?

Reviewer #2: Yes

Reviewer #3: (No Response)

3. Has the statistical analysis been performed appropriately and rigorously? 

Reviewer #2: I Don't Know

Reviewer #3: (No Response)

4. Have the authors made all data underlying the findings in their manuscript fully available?

Reviewer #2: Yes

Reviewer #3: (No Response)

5. Is the manuscript presented in an intelligible fashion and written in standard English?

Reviewer #2: Yes

Reviewer #3: (No Response)

6. Review Comments to the Author

Reviewer #2: No evidence of duplicate publication was found in the search, and the authors have strictly adhered to research and publication ethics.Thanks to the efforts of the authors, the manuscript is well edited and ready for publication.

Reviewer #3: (No Response)

7. PLOS authors have the option to publish the peer review history of their article (what does this mean?). If published, this will include your full peer review and any attached files.

Reviewer #2: **Yes: **Mohammad Hosein Hayavi-Haghighi

Reviewer #3: No

---

## [Editor Report · Acceptance letter]

29 Apr 2024

PONE-D-23-38333R2 

PLOS ONE

Dear Dr. Chereka, 

I'm pleased to inform you that your manuscript has been deemed suitable for publication in PLOS ONE. Congratulations! Your manuscript is now being handed over to our production team.

Kind regards, 

on behalf of

Dr., Jahanpour Alipour 

Academic Editor

PLOS ONE